# Neighborhood Violent Crime and Perceived Stress in Pregnancy

**DOI:** 10.3390/ijerph17155585

**Published:** 2020-08-03

**Authors:** Megan M. Shannon, Jane E. Clougherty, Clare McCarthy, Michal A. Elovitz, Max Jordan Nguemeni Tiako, Steven J. Melly, Heather H. Burris

**Affiliations:** 1Perelman School of Medicine, University of Pennsylvania, Philadelphia, PA 19104, USA; Megan.Shannon@pennmedicine.upenn.edu; 2Department of Environmental and Occupational Health, Dornsife School of Public Health, Drexel University, Philadelphia, PA 19104, USA; jec373@drexel.edu; 3Maternal and Child Health Research Center, Department of Obstetrics and Gynecology, University of Pennsylvania Perelman School of Medicine, Philadelphia, PA 19104, USA; Clare.Mccarthy@pennmedicine.upenn.edu (C.M.); elovitz@pennmedicine.upenn.edu (M.A.E.); 4Yale School of Medicine, Yale University, New Haven, CT 06510, USA; max.tiako@yale.edu; 5Urban Health Collaborative, Dornsife School of Public Health, Drexel University, Philadelphia, PA 19104, USA; sjm389@drexel.edu; 6Division of Neonatology, Department of Pediatrics, Children’s Hospital of Philadelphia, Philadelphia, PA 19104, USA; 7Department of Pediatrics, Perelman School of Medicine, University of Pennsylvania, Philadelphia, PA 19104, USA

**Keywords:** perceived stress, neighborhood, census tract, violent crime, pregnancy

## Abstract

Stress has been shown to adversely affect pregnancy outcomes. Neighborhood crime rates may serve as one publicly available social determinant of health for pregnancy studies that use registry or electronic health record datasets in which individual-level stress data are not available. We sought to determine whether neighborhood violent crime incidents were associated with measured perceived stress in a largely minority, urban pregnancy cohort. We performed a secondary analysis of the 1309 Philadelphia residents participating in the *Motherhood and Microbiome* cohort (*n* = 2000) with both neighborhood violent crime and Cohen’s Perceived Stress Scale (PSS-14) data. Generalized linear mixed models accounting for confounding variables and geographic clustering demonstrated that, regardless of race, women with the highest quartile of neighborhood violent crime had significantly elevated odds of high stress compared to women with lower crime. We also found that Black women were more likely to have both the highest quartile of neighborhood violent crime and high stress than non-Black women. Overall, this study demonstrates that neighborhood violent crime is associated with perceived stress in pregnancy. Given disparate exposure to crime and prenatal stress by race, future work is warranted to determine whether urban neighborhood violence and/or stress reduction strategies would improve birth outcome racial disparities.

## 1. Introduction

Adverse pregnancy outcomes disproportionately affect women of color and poor women in the U.S. and are major predictors of infant morbidity and mortality [1]. While numerous studies have attempted to determine the etiology of these persistent birth outcome disparities, a single unifying explanation has not been elucidated [2]. Studies examining the aggregate, combined influence of multiple factors are likely needed to better explain and resolve racial health disparities [3].

Prenatal maternal stress is one factor that has been associated with increased risk of adverse birth outcomes, including preterm birth and low birth weight [4,5,6]. Wadhwa et al. demonstrated that pregnant women who reported higher levels of life event stress, measured using Cohen’s Perceived Stress Scale (PSS) among other standardized questionnaires, had increased risk of delivering low-birth-weight neonates [6]. The study also found that elevated pregnancy anxiety was associated with lower gestational age at birth [6]. These findings are consistent with other reports of positive associations between high PSS scores and risks of low birth weight and preterm birth [7,8]. However, while maternal perceived stress has been demonstrated to significantly impact birth outcomes, individual-level stress data are not always readily available, particularly in large population-based registries.

In contrast, neighborhood violent crime rates may be publicly available and easily accessible. Research has shown that neighborhood violence is a risk factor for poor mental health [9,10]. Many studies in the literature have examined the effect of perceived neighborhood violence on stress levels in urban, nonpregnant cohorts and have shown that high levels of perceived neighborhood violence and crime are associated with increased psychological distress and depression [11,12,13,14]. A limited number of studies involving pregnant cohorts have also supported this association [15,16]. Further, neighborhood violent crime is highly associated with residents’ perception of neighborhood violence, whereas other forms of crime, such as property crime, are less strongly associated [17].

Given that large registry databases will likely be needed to identify contributors to birth outcome disparities, especially when effects may be subtle, the use of publicly available data on neighborhood violence may serve as one potential proxy measure for maternal stress. To date, however, studies examining the association between neighborhood-level violent crime and measured perceived stress in pregnant, urban cohorts have been lacking in the literature. Therefore, we aimed to investigate whether publicly available neighborhood violence rates—defined as incidents of aggravated assault with and without a firearm, robbery with and without a firearm, rape, other assaults, and homicide—were associated with measured perceived stress in a prospective pregnancy cohort.

## 2. Materials and Methods 

### 2.1. Study Design

We conducted a secondary analysis of a prospective cohort study, *Motherhood and Microbiome* (*n* = 2000) [18,19]. The methods of the original study were previously published [18]. Briefly, women with singleton gestation pregnancies receiving prenatal care at the University of Pennsylvania enrolled between December 2013 and February 2017 after providing written, informed consent. Exclusion criteria included major fetal anomalies, HIV seropositive status, history of organ transplant, chronic steroid use, or enrollment into the study during a previous pregnancy.

Of the 2000 initial participants, 1813 women completed Cohen’s Perceived Stress Scale (PSS-14) in the first visit at 16–20 weeks of gestation [20]. Of these women, 1309 lived in Philadelphia, PA, where community violent crime data were available, and comprised the analytic sample for this analysis (Figure 1). The Institutional Review Board of the University of Pennsylvania approved this study (IRB #818914). 

### 2.2. Exposure—Neighborhood Violent Crime

Publicly available violent crime data from the City of Philadelphia were downloaded and curated at the Urban Health Collaborative at the Drexel University Dornsife School of Public Health [21]. Violent crime was defined as aggravated assault with and without a firearm, robbery with and without a firearm, rape, other assaults, and homicide. Using violent crime data from 2015, we then calculated annual census tract incidents per square kilometer in 212 census tracts. We divided violent crime incidents into quartiles among cohort participants (as opposed to quartile of tract crime incidents) and defined high exposure as the highest quartile of violent crime among participants (>352.5 violent crimes/square kilometer per year). Spatial analysis was conducted and maps were created using ArcGISPro 2.5.0. (Esri, Redlands, CA, USA)

### 2.3. Outcome—Perceived Stress

Participants completed Cohen’s Perceived Stress Scale (PSS-14) during three clinical visits. These visits occurred between 16–20 (visit 1), 20–24 (visit 2), and 24–28 (visit 3) weeks of gestation. As previously done by us and others, we classified women as having high stress levels if their score was ≥30, which is the 80th percentile in the *Motherhood and Microbiome* cohort [22,23]. We chose to focus this analysis on the first visit (16–20 weeks) because most women had stress data from that visit and because the scores had moderate-to-strong correlations at the three time points (r = 0.64–0.74, *p* < 0.0001).

### 2.4. Covariate Ascertainment

We considered several variables as potential confounders of the association between neighborhood violence and stress, including self-reported race, age, parity, education level, insurance status, marital status, and comorbid conditions such as pregestational diabetes and chronic hypertension. Race was dichotomized as Black and non-Black due to small numbers of non-Black participants. Neighborhood characteristics, including percent foreign-born, poverty, and high school graduation rates, were also assessed.

### 2.5. Statistical Analysis

Bivariate analysis of cohort demographics (race, age, parity, education level, insurance status, marital status, and comorbid conditions) and neighborhood characteristics (percent foreign-born, poverty, high school graduation rates) were analyzed in association with neighborhood violent crime incidents and maternal stress levels. We used t-tests and chi-square tests to analyze unadjusted associations of these variables with maternal stress levels and neighborhood violence, as appropriate. We examined scatter plots and correlations between neighborhood violent crime and stress. We considered *p*-values < 0.05 significant.

For our primary analysis, we used generalized linear mixed models (Proc Glimmix, SAS 9.4, Carey, NC, USA) to model the unadjusted and adjusted associations (odds ratios and 95% confidence intervals) of high neighborhood violent crime with high perceived stress, clustering by census tract. Covariates, either alone or in conjunction with others, that substantially altered the association of crime with stress by more than ~10% were retained in the adjusted model, while variables that did not confound the association were dropped to achieve parsimony. Furthermore, we performed models stratified by race. Finally, as a sensitivity analysis, we explored whether a different measure of crime at a finer scale (counts within 400 m buffers surrounding each participant’s residence) were similarly associated with perceived stress.

## 3. Results

Annual violent crime incidents per square kilometer across Philadelphia are shown in Figure 2. Demographic characteristics of study participants in relation to neighborhood crime and individually measured perceived stress scores are presented in Table 1. Black women were more than four times more likely to live in neighborhoods with high violent crime levels (31.7%) than non-Black women (6.7%). Women in the highest quartile of neighborhood violent crime were more likely to be young, publicly insured, and have less education. Black participants were three times more likely to report high levels of stress (PSS-14 ≥ 30) (26.3%) than non-Black participants (8.6%). Women reporting high levels of stress were also more likely to be young, publicly insured, and less educated. Because we only had access to violent crime incidents within the City of Philadelphia, we needed to exclude suburban women in the *Motherhood and Microbiome* cohort from this analysis (sociodemographic characteristics of included and excluded women are compared in Appendix A). As expected, excluded women were more likely to be non-Black and privately insured. They also had lower prevalence of high stress.

Participants lived in 212 census tracts in Philadelphia. Women with the highest incidents of neighborhood violent crime (defined by highest quartile of tract incidents among participants) lived in 44 tracts (20.8% of the tracts). Compared to census tracts with lower violent crime levels, tracts with higher levels had significantly lower proportions of foreign-born residents, higher poverty rates, and a lower proportion of adults with at least a high school education (Table 2).

Scatter plots revealed that the association between neighborhood violent crime and perceived stress was largely linear, but the correlation was modest (r = 0.14). Pregnant women with the highest quartile of neighborhood violent crime had higher mean PSS-14 scores at 16–20 weeks of gestation (mean 24.8, SD 8.0) than women with lower exposures (mean 22.7, SD 7.8) (*p* < 0.0001). Women with high neighborhood violent crime were more likely to have a high PSS-14 score (≥30) (29.5%) compared to women with lower neighborhood crime (18.8%) (Table 3). Adjustment for race, age, and insurance status, as well as neighborhood poverty and foreign-born rates, attenuated the association between high violence and high perceived stress, but it remained significant (adjusted OR 1.38, 95% CI: 1.004–1.89) (Table 4). No additional confounding was noted with education, marital status, comorbid conditions (diabetes, hypertension), parity, or neighborhood education. 

Black women were more likely than non-Black women to have both elevated neighborhood violent crime incidents and high perceived stress; thus, we performed stratified analyses by race. Among Black women, high neighborhood violent crime was significantly associated with high perceived stress in both unadjusted and adjusted models (unadjusted OR 1.40, 95% CI: 1.03–1.90; adjusted OR 1.41, 95% CI: 1.01–1.95), with little evidence of confounding by covariates (Table 4). There were just three non-Black women with both high neighborhood violent crime and high stress levels, leading to wide confidence intervals but point estimates that were similar (unadjusted OR 1.57, 95% CI: 0.44–5.62; adjusted OR 1.27, 95% CI: 0.31–5.17). 

In the sensitivity analysis, violent crime incidents in census tracts were highly correlated with violent crime counts within 400 m buffers of participants’ residences (r = 0.81). When we analyzed annual violent crime counts within 400 m buffers of individuals’ residences, results were comparable to the primary analysis. Per interquartile range of violent crime counts, women had elevated odds of high stress in unadjusted models (OR 1.57, 95% CI: 1.30–1.90). This association was attenuated in adjusted models, but the point estimate remained similar to the primary analysis (adjusted OR 1.20, 95% CI: 0.96–1.49).

## 4. Discussion

We found that neighborhood violent crime was associated with high levels of perceived stress in pregnancy. This association was robust to adjustment for confounders and was also present when restricted to Black women. This is particularly important given that elevated maternal stress has been shown to increase risk of adverse birth outcomes that disproportionately affect Black families, such as preterm birth, small for gestational age, and low birth weight [1,3,4,5,6,24,25,26].

There are two potential implications of our findings. First, crime may represent a social determinant of health that is publicly available and could be useful in epidemiologic studies that use large registries and electronic health records where individually measured stress is not available. However, since there was only a modest correlation between neighborhood violent crime and stress, violent crime alone is insufficient to serve as a proxy for stress. Second, there is evidence that there are neighborhood interventions that may reduce stress [27,28,29]; as such, thoughtful urban planning may reduce stress in pregnancy, which could improve perinatal outcomes. 

Our findings are consistent with a study by Dustmann et al., which analyzed two longitudinal survey results of English adults (*n* = approximately 9000 and 12,000) and found that overall local crime rates had a significant impact on the mental distress of urban residents [30]. This association was particularly strong in women [30]. Within pregnant populations, Barcelona de Mendoza et al. demonstrated that pregnant women who reported higher levels of perceived neighborhood violence were more likely to have depressive symptoms [16]. Previous research has also shown that census tract violent crime is strongly correlated with residents’ perception of crime [17]. However, not all studies confirm the association between neighborhood violence and stress, and some have instead found associations with perceived crime rates but not with crime data maintained by municipalities from law enforcement agencies. For example, an analysis of participants with current or past drug use disorder in Baltimore, Maryland (*n* = 786), found that it was perception of neighborhood disorder that correlated with depressive symptoms, rather than quantified crime rates [31]. In this study, neighborhoods were analyzed via nested neighborhood block groups, rather than via census tract data as in our study. Similarly, within an urban, pregnant cohort (*n* = 101), Giurgescu et al. found that perceptions (as opposed to city-maintained data) of neighborhood conditions such as crime, social disorder, and physical disorder in half-mile circular buffers around participants’ homes were associated with maternal psychological distress [32]. Discordance between perceptions of crime and city-maintained crime statistics could be due to two potential reasons. One possibility is that perceptions could be different from reality. The second possibility is that the city-maintained crime incidents are not representative of true crime incidents due to lack of reporting. It is possible that actual crime incidents and city-maintained incidents were closely matched in our study, leading to a positive association with stress. We did not ask women about perceptions of violent crime. To our knowledge, our study is the first to find an association between city-maintained neighborhood violent crime incidents and stress levels in a pregnant cohort. 

Our findings are important due to recent evidence that environmental interventions, such as greening of vacant lots, reduced neighborhood crime rates within the city of Philadelphia, PA, USA. [27,28]. Additionally, South et al. conducted a cluster randomized trial (*n* = 342) which found that Philadelphia residents whose neighborhoods were randomized to vacant lot greening reported improved mental health [29]. Other neighborhood interventions such as blight remediation and public-housing modifications have also been shown to decrease neighborhood crime rates [33,34,35,36]. Thus, given that there is an association between neighborhood violence and stress levels in pregnant women, it is possible that neighborhood efforts to reduce violent crime, such as environmental interventions, may improve mental health among pregnant women and could, thereby, lead to improved birth outcomes.

Within our study, we also found that Black women living in Philadelphia were significantly more likely to live in neighborhoods with high levels of violent crime and were more likely to report high perceived stress than non-Black women. This racial disparity is consistent with other reports in the literature [37,38]. Additionally, we observed little evidence of confounding by other socioeconomic variables, including insurance status, on the relationship between neighborhood violence and stress among Black women. In contrast, we observed some attenuation of the association between violence and stress among non-Black women in adjusted models, although the sample size in this subgroup was small and estimates were imprecise. It remains possible that other unmeasured confounders and/or modifiers are responsible for the relationship between violence and chronic stress among Black women, such as racial discrimination and other consequences of structural racism [39,40,41,42]. We did not assess for racial discrimination in our study, but further study to understand the complex interplay of neighborhood exposures and interpersonal discrimination on stress among Black women is warranted. 

Strengths of our study included its integration of individual- and neighborhood-level variables. However, our study also had limitations. Our findings could have resulted from unmeasured confounding, as we do not know whether area-level violence or some other factor was leading to increased stress among women in neighborhoods with high levels of violent crime. We did not ascertain employment status, level of perceived social support, prior spontaneous abortions, desired versus undesired pregnancy, or history of mental illness, any of which might have attenuated our results. Women were also not asked whether they had experienced neighborhood violence. Further, our study utilized crime data by census tract, which does not always approximate self-defined or perceived neighborhoods [43,44]. The use of census tract crime incidents could have also led to exposure misclassification due to variability of crime incidents within tracts; however, in the sensitivity analysis, use of finer-scale crime counts (within 400 m buffers) revealed similar findings to our primary analysis. Furthermore, there are inherent limitations in utilizing publicly available violent crime data, such as possible under-reporting of violent crimes such as rape and domestic violence, with reported rates potentially varying depending upon the geographic or demographic characteristics of a population. Moreover, our study population was restricted to women living in the City of Philadelphia. Although this analytic cohort differed significantly from the larger *Motherhood and Microbiome* cohort, it was likely representative of the urban population within Philadelphia, although not necessarily generalizable to suburban women. Finally, our study highlights that it is possible to link hospital data to crime data to better identify ways to improve the health and safety of populations, a practice that is beginning to see use in other fields such as criminology [45,46].

## 5. Conclusions

In conclusion, we found that, regardless of race, neighborhood violent crime was associated with high levels of perceived stress in an urban, pregnant cohort. Future work focused on assessing the generalizability of this association is needed to ensure that neighborhood violent crime incidents can be a useful, publicly available social determinant of health for larger registry-based studies. Further, given the finding of disparate exposure to crime and prenatal stress by race within our study, future work is warranted to determine whether urban neighborhood violence and/or stress reduction strategies would alleviate racial disparities in birth outcome.

## Figures and Tables

**Figure 1 ijerph-17-05585-f001:**
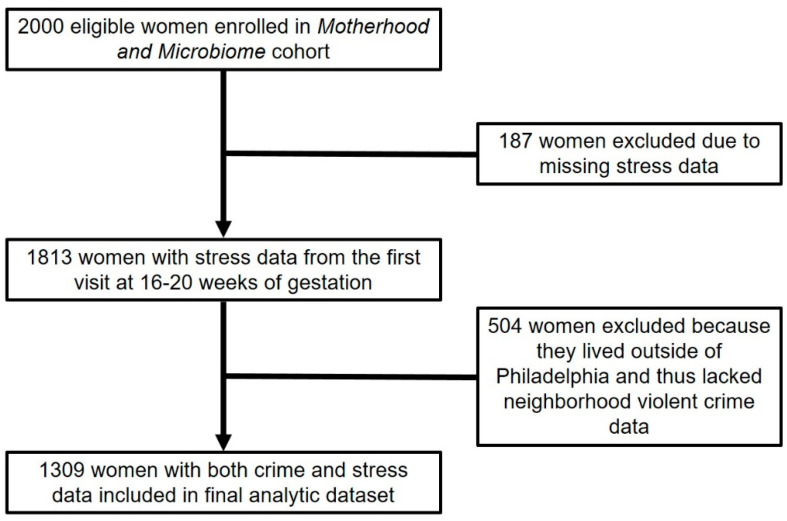
Flowchart of study subjects included and excluded in the analytic dataset.

**Figure 2 ijerph-17-05585-f002:**
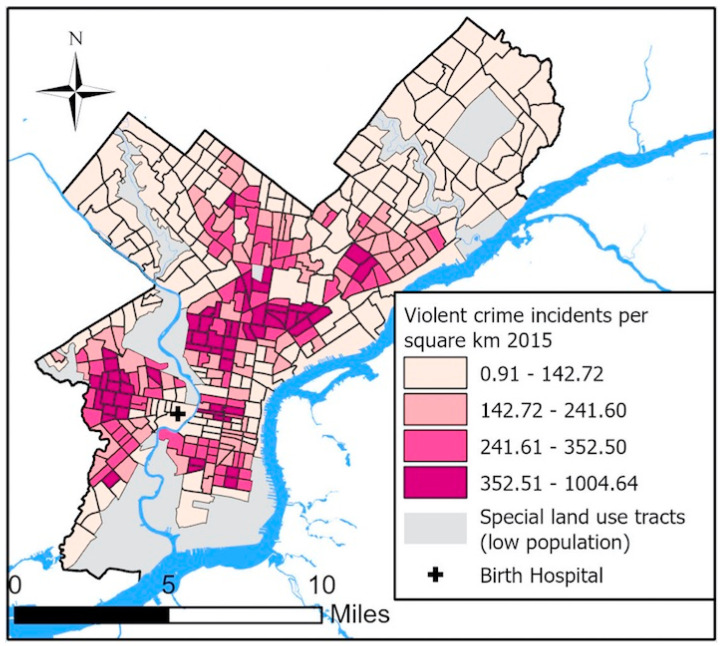
Neighborhood violent crime across Philadelphia, 2015.

**Table 1 ijerph-17-05585-t001:** Neighborhood violent crime and perceived stress among 1309 pregnant women in Philadelphia participating in the *Motherhood and Microbiome*, by sociodemographic and health characteristics.

	Annual Census Tract Violent Crime Incidents per Square Kilometer		Perceived Stress PSS-14 Score at 16–20 Weeks of Gestation	
	Low (≤352.5), Quartiles 1–3 (*n* = 984)	High (>352.5), Quartile 4 (*n* = 325)	Low (<30) (*n* = 1028)	High (≥30) (*n* = 281)
**Characteristics**	***n* (%)**	***n* (%)**	***n* (%)**	***n* (%)**
**Race**				
Black	649 (68.3)	301 (31.7)	700 (73.7)	250 (26.3)
Non-Black	335 (93.3)	24 (6.7)	328 (91.4)	31 (8.6)
**Age**				
<25	291 (66.6)	146 (33.4)	312 (71.4)	125 (28.6)
25–34	530 (78.1)	149 (21.9)	550 (81.0)	129 (19.0)
≥35	163 (84.5)	30 (15.5)	166 (86.0)	27 (14.0)
**Parity**				
0	454 (78.3)	126 (21.7)	463 (79.8)	117 (20.2)
>0	530 (72.7)	199 (27.3)	565 (77.5)	164 (22.5)
**Insurance status**				
Medicaid/uninsured	542 (68.4)	250 (31.6)	576 (72.7)	216 (27.3)
Private	442 (85.5)	75 (14.5)	452 (87.4)	65 (12.6)
**Maternal education ^1^**				
High school graduate or lower	543 (67.3)	264 (32.7)	592 (73.4)	215 (26.6)
College graduate or higher	363 (90.1)	40 (9.9)	368 (91.3)	35 (8.7)
**Marital status**				
Married	307 (88.7)	39 (11.3)	315 (91.0)	31 (9.0)
Single/separated/divorced	677 (70.3)	286 (29.7)	713 (74.0)	250 (26.0)
**Pregestational diabetes ^1^**				
Yes	22 (84.6)	4 (15.4)	17 (65.4)	9 (34.6)
No	958 (75.0)	320 (25.0)	1007 (78.8)	271 (21.2)
**Chronic hypertension ^1^**				
Yes	50 (69.4)	22 (30.6)	53 (73.6)	19 (26.4)
No	932 (75.6)	301 (24.4)	972 (78.8)	261 (21.2)

^1^ Education data missing for 99 women, diabetes data missing for 5 women, hypertension data missing for 7 women.

**Table 2 ijerph-17-05585-t002:** Census tract characteristics (212 Philadelphia census tracts) by annual census tract violent crime incidents per square kilometer, 2015.

Census Tract Characteristics	Low (≤352.5) *n* = 168 Census Tracts	High (>352.5) *n* = 44 Census Tracts	*p*
	Percent (SD)	Percent (SD)	
Foreign-born	10.4 (8.4)	7.7 (7.6)	0.05
Poverty	26.4 (13.7)	40.7 (12.0)	<0.0001
High school or higher education level	83.7 (9.2)	76.4 (11.5)	0.0003

**Table 3 ijerph-17-05585-t003:** Neighborhood violent crime incidents per square kilometer and perceived stress (*Motherhood and Microbiome* cohort, *n* = 1309).

Stress Levels	Low (≤352.5) *n* = 168 Census Tracts	High (>352.5) *n* = 44 Census Tracts	*p*
	*n* (%)	*n* (%)	
Low (PSS-14 < 30)	799 (81.2)	229 (70.5)	<0.0001
High (PSS-14 ≥ 30)	185 (18.8)	96 (29.5)	

PSS-14, Cohen’s Perceived Stress Scale.

**Table 4 ijerph-17-05585-t004:** Unadjusted and adjusted ^1^ associations of the highest quartile of neighborhood violent crime and high levels of individual perceived stress (PSS-14 ≥ 30), *Motherhood and Microbiome* cohort, *n* = 1309.

Models	OR	(95% CI)
**All participants (*n* = 1309)**		
Unadjusted	1.82	(1.35–2.45)
Adjusted	1.38	(1.004–1.89)
**Race-stratified models**		
Black (*n* = 950)		
Unadjusted	1.40	(1.03–1.90)
Adjusted	1.41	(1.01–1.95)
Non-Black (*n* = 359)		
Unadjusted	1.57	(0.44–5.62)
Adjusted	1.27	(0.31–5.17)

^1^ Adjusted for age, race, insurance, neighborhood percent poverty, and foreign-born. Stratified models similarly adjusted (except race was not included).

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
