# Peer review of "Neighborhood Violent Crime and Perceived Stress in Pregnancy"

_ijerph, 2020, doi:10.3390/ijerph17155585_

Round 1

Reviewer 1 Report

Authors present quite an interesting topic about crime and pregnancy stress. Nevertheless, there are some points that need to be improved:

  1. Introduction

1.1 What is the reason to write about black infant deaths? I am afraid that there is a lot of information in the introduction about black deaths that it is not relevant.

If you keep the information, it is important to reestructure the introduction.

1.2 When you talk about pregnancy and prenatal stress, why don´t include some studies where prenatal stress has been measured with the same instrument as yours (PSS-14)? There are some recent papers about perceived stress in pregnancy and negative consequences in maternal and infant health.

1.3 Why don´t write about how perceived stress in pregnancy could have an impact on fetal and newborn infant health? (i. e. newborn infant reactivity to stress, newborn infant cortisol, etc.)

1.4 Please, reorganize introduction so you write first about neighborhood and crime rates, then about pregnancy, and then the relationship between them.

2. Methods are clearly written, congrats!

3. Results:

3.1 There are some variables that can influence PSS scores and they have not been into account, please, include them, as confounding variables in the analyses and in descriptive information:

Marital status

Parity

Education

Previous miscarraiges

Number of children

4. Discusion

Line 233, yo stated that "Thus, it is possible that neighborhood efforts to reduce violent crime may improve mental health among pregnant women, and could thereby lead to improved birth outcomes."

It is needed to explain the possible conection between reducing violent crime and improvind birth outcomes. 

Line 237, please specify the exact location where the study has been performed, otherwise it seems that black women are more likely to live in places with high crime rates across the entire USA.

Reviewer 2 Report

The authors examined the relationship between neighborhood violent crime with perceived stress in pregnancy. In sum, the paper is well written and structured. However, it seems there are some points in the manuscript that needs improvement to be published in this journal. I tried to help with some straightforward comments below. My overall recommendation is to accept after revision.

1- Statistical analysis – usually, neighborhood violent crime models consider spatial autocorrelation because crime is highly concentrated in the space. Please, inform us why did you not mention this fact. Consider checking the spatial dependency of your variables.

2- I suggest you increase your map (Figure 2).

3- I really liked the second implication of your results (line 198). You could explore it more. Actually, crime data could be useful in epidemiologic studies (e.g. this research), on the other hand, medical data also could be useful to criminal studies. Current criminology research is pointing out how crime is underreported (this paper well recognize that) and needs other sources such as epidemiologic data for a better picture. Please take a look at these articles:

Hibdon, J., Telep, C. W., & Groff, E. R. (2017). The concentration and stability of drug activity in Seattle, Washington using police and emergency medical services data. Journal of quantitative criminology33(3), 497-517.

Melo, S. N., Boivin, R., & Morselli, C. (2020). Spatial dark figures of rapes:(In) Consistencies across police and hospital data. Journal of Environmental Psychology68, 101393.

4- "We found that Black women were more likely to have both the highest quartile of neighborhood violent crime and higher stress than non-Black women". I believe this is an important result that you need to incorporate in the conclusion.

Reviewer 3 Report

The text requires supplementation and serious reflection
on the method.
A clearly defined research gap is missing. The Authors emphasize that they examined the
impact of crime on stress in pregnant women.
I don't think that this problem is new and revealing.
Each type of crime is a stress factor.
Regardless of whether this applies to pregnant women or not.
There is no background for the research assumptions -
it is possible that the socio-economic one indicates
such a drastic division of women. He is not convincing
at the moment. Ethically worries most in the way of
adopted names (Black women, non-Black women.
The abstract does not reflect the elements referred
to in the discussion and ultimately in the conclusions.
In the discussion, the Authors point out that: "Our findings
are consistent with a study by Dustmann
et al.,
which analyzed two longitudinal
survey results of English
adults (n = approximately 9,000 and 12,000) and found that
overall local
crime rates had a significant impact on the
mental distress of urban residents [28]. This association
was particular
ly strong in women" - line 203-205
Speaking of women of the city of Philadelphia,
how do we suddenly start talking about the people of England?
In this case, paper requires authors to think deeply
about the logic of building its structure.
The situation is similar in the case of neighborhood
crime with greening interventions(their impact on the
sense of security
among the respondents)- line: 196-198

Other remarks: Very poor conclusions that do not highlight the problems
and results obtained during the research.
There is no connection between the place of residence
of pregnant women and the location of crimes.
It seems that the city area is not accurate enough
to determine such relationships.

Figure 2 - pay attention to the legend and improve
class boundaries.
From line 103 to line 126 - please note the text editing.            

Round 2

Reviewer 1 Report

.

Author Response

Thank you for your prior comments.

Reviewer 3 Report

I maintain my view of the lack of evidence that pregnant women
respond differently to stress than the rest of the population.
I have great ethical concerns regarding the thesis
presented by the Authors - concerning the division of
women by colour
of their skin. The method correction and explanations
don't convince me that the above assumption of the Authors
are correct.

The comment on the method improvement requirements
is still valid.

The text requires supplementation and serious reflection on the method. A clearly defined research gap is missing. The Authors emphasize that they examined the impact of crime on stress in pregnant women. I don't think that this problem is new and revealing. Each type of crime is a stress factor. Regardless of whether this applies to pregnant women or not.

The text requires strong links with the literature within
the scope of the indicated theses. It is necessary to clarify the connection between the way of land use and the results of the research. The topic was not described but only supported by the literature. Out of nowhere, the final paragraphs of the text mention urban and green space, when the method does not mention it.
